# Targeting KRAS in Pancreatic Ductal Adenocarcinoma: The Long Road to Cure

**DOI:** 10.3390/cancers15205015

**Published:** 2023-10-17

**Authors:** Victor Hugo Fonseca de Jesus, Maria Cecília Mathias-Machado, João Paulo Fogacci de Farias, Marcelo Porfirio Sunagua Aruquipa, Alexandre A. Jácome, Renata D’Alpino Peixoto

**Affiliations:** 1Department of Gastrointestinal Medical Oncology, Oncoclínicas, Florianópolis 88015-020, Brazil; 2Department of Gastrointestinal Medical Oncology, Oncoclínicas, São Paulo 04538-132, Brazil; 3Department of Gastrointestinal Medical Oncology, Oncoclínicas, Rio de Janeiro 22775-003, Brazil; 4Department of Gastrointestinal Medical Oncology, Oncoclínicas, Belo Horizonte 30360-680, Brazil

**Keywords:** pancreatic, cancer, KRAS, mutation, inhibitor

## Abstract

**Simple Summary:**

Pancreatic ductal adenocarcinoma is one of the deadliest malignancies in humans. Despite advances in systemic therapy, prognosis still remains poor. However, recent discoveries in the main biological event in pancreatic carcinogenesis, the *KRAS* mutation, have broadened our understanding of pancreatic cancer and opened a window of opportunity. Indeed, inhibitors of KRAS signaling are believed to represent a major step toward more active treatments against this disease. In this review, we describe the latest findings regarding KRAS in pancreatic ductal adenocarcinoma, along with updated preclinical and clinical data on KRAS-targeted therapy.

**Abstract:**

Pancreatic ductal adenocarcinoma (PDAC) remains an important cause of cancer-related mortality, and it is expected to play an even bigger part in cancer burden in the years to come. Despite concerted efforts from scientists and physicians, patients have experienced little improvement in survival over the past decades, possibly because of the non-specific nature of the tested treatment modalities. Recently, the discovery of potentially targetable molecular alterations has paved the way for the personalized treatment of PDAC. Indeed, the central piece in the molecular framework of PDAC is starting to be unveiled. *KRAS* mutations are seen in 90% of PDACs, and multiple studies have demonstrated their pivotal role in pancreatic carcinogenesis. Recent investigations have shed light on the differences in prognosis as well as therapeutic implications of the different *KRAS* mutations and disentangled the relationship between KRAS and effectors of downstream and parallel signaling pathways. Additionally, the recognition of other mechanisms involving KRAS-mediated pathogenesis, such as KRAS dosing and allelic imbalance, has contributed to broadening the current knowledge regarding this molecular alteration. Finally, KRAS G12C inhibitors have been recently tested in patients with pancreatic cancer with relative success, and inhibitors of KRAS harboring other mutations are under clinical development. These drugs currently represent a true hope for a meaningful leap forward in this dreadful disease.

## 1. Introduction

Pancreatic ductal adenocarcinoma (PDAC) is currently the third most frequent cause of cancer-related death in the USA [1], and recent studies foresee an increase in the epidemiological burden of the disease in the years to come. Most patients with PDAC present with synchronous metastatic disease, and the prognosis in this setting is poor [2,3]. Even with recent developments in systemic therapy [4,5,6], average survival is less than a year. These hurdles fostered the study of the molecular mechanisms underpinning PDAC initiation and progression in an attempt to improve treatment outcomes.

Basic science studies have identified *KRAS* as the most frequently mutated gene in PDAC and the culprit for many of the biological mechanisms that explain its aggressive behavior and resistance to therapy [7]. Initial efforts to directly or indirectly target *KRAS* in PDAC were disappointing. Now, a new generation of drugs designed with state-of-the-art chemical techniques and novel biological insights gives us hope that tackling *KRAS*-mutated PDAC is finally possible.

Building on previous reviews and recent studies on the topic, we performed a narrative review of the current and emerging treatment strategies against *KRAS*-mutated PDAC. We searched PubMed using the terms “pancreatic AND cancer AND KRAS AND mutation” and meeting abstracts from the American Society of Clinical Oncology (ASCO), the European Society of Medical Oncology (ESMO), and the American Association for Cancer Research (AACR) annual meetings from 2015 to 2023.

## 2. KRAS as a Member of Intracellular Signaling Pathways

In humans, the RAS superfamily comprises more than 150 functionally related small proteins with GTP hydrolyzing activity (GTPase), divided structurally and functionally into five families (RAS, RHO, ARF, RAN, and RAB) [8,9]. While the RAS family also includes other important members of different signaling pathways (such as Rap, RRAS, Ral, and Rheb), the most important members of the RAS family in humans are KRAS, NRAS, and HRAS [10]. They are key players in the regulation of many intracellular physiological mechanisms, such as proliferation, migration, survival, and differentiation and were first discovered in studies of the acute transforming rat sarcoma viruses Ha-SV (Harvey Sarcoma Virus, 1964) and Ki-SV (Kirsten Sarcoma Virus, 1967) and of the neuroblastoma cell line SK-N-SH (1983) [8]. In pancreatic ductal adenocarcinoma (PDAC), *NRAS* (0.65%) and *HRAS* (0.2%) mutations are exceedingly rare [11,12]. Therefore, we will focus on the role of *KRAS* mutations in the pathogenesis, prognosis, and treatment of PDAC.

The protooncogene *KRAS* (chromosome 12p12.1) encodes two different 21 kDa KRAS proteins (KRAS4A and KRAS4B) through alternative splicing of the last exon (exon 4)—Figure 1 [13]. These two variants differ mainly in their C-terminal hypervariable region (HVR; residues 167–189) but also in residues 151, 153, 165, and 166 [14]. Also, these two variants display different post-translational modifications (see below) and have non-completely overlapping downstream effectors, possibly due to differences in membrane sublocation [15]. In normal tissue and in pancreatic cancer cell lines, KRAS4B is the dominant isoform [16]. However, KRAS4A also has oncogenic properties and, indeed, was the first splice variant identified as it represents the transforming gene of Ki-SV [17].

KRAS4A and KRAS4B share a conserved GTPase domain (G domain; residues 1–166) and differ mainly in the HVR—Figure 2 [14]. The G domain is composed of six beta-strands surrounded by five alpha-helices, configuring an N-terminal effector lobe and a C-terminal allosteric lobe [18]. Key enzymatic structures are located in the N-terminal effector lobe, such as the P-loop (residues 10–17), switch I (residues 30–38) and switch II (residues 59–76) regions [19]. This is specifically important because almost all missense mutations affecting the *KRAS* gene in PDAC occur in hotspots located in the P-loop (G12 or G13) or switch II (Q61) coding regions and interfere with the dynamics of the switch I and switch II regions (see below) [20].

Proper KRAS functioning depends on its trafficking to the inner surface of the plasma membrane. This process is regulated by post-translational modifications of KRAS4A and KRAS4B, which are partially different—Figure 2 [19]. Both isoforms undergo farnesylation of the cysteine residue present in the C-terminal CAAX sequence (where C means cysteine, A means aliphatic amino acid, and X means any amino acid) by farnesyltrasferase (FTase). After that, RAS-converting enzyme 1 (RCE1) cleaves the AAX sequence, and the isoprenylcysteine carboxyl methyltransferase (ICMT) transfers a carboxymethyl moiety to the farnesylated cysteine. However, trafficking between membranes requires additional steps that differ between KRAS4A and KRAS4B. KRAS4A undergoes palmitoylation before trafficking to the plasma membrane. In the case of KRAS4B, a poly-lysine sequence in the HVR interacts with negatively charged lipids in the plasma membrane, leading to membrane attachment. After these post-translational modifications, phosphodiesterase-delta (PDEδ) binds to RAS (mainly KRAS4B) and shuttles it to the plasma membrane [21]. Importantly, other post-translational modifications might target KRAS isoforms to distinct microenvironments in the plasma membrane and enable the trafficking of KRAS to endomembranes.

Once attached to the plasma membrane, individual GTP-bound RAS proteins seem to cluster into dimers or even higher-order nanoclusters (5 to 10 monomers) [19]. However, it is not currently known which interface (α3–α4 helix, α–α5 helix, or β-sheet) drives the assembly of these multi-RAS complexes or the role of such complexes in RAS signaling transduction [22]. Moreover, α–α5 helixes do not have intrinsic dimerization capacity and dimerization-disrupting mutations in this region have not demonstrated impaired self-association of RAS or signaling transduction [22].

Effector protein biding is controlled by the two switch regions. As a GTPase, KRAS cycles over an active configuration when bound to GTP and an inactive one when bound to GDP [19]. The KRAS enzymatic conformational change has been compared to a loaded-spring mechanism: when KRAS binds GTP, the γ-phosphate of GTP interacts with the threonine 35 residue in switch I and the glycine 60 residue in switch II, rendering KRAS in the active effector-binding configuration. When GTP is hydrolyzed, the GDP-bound RAS changes to an inactive configuration in which both switch regions are relaxed. Of note, the affinity for effectors seems to be dictated by the conformation of RAS rather than by the nucleotide state itself [23].

Given the very slow off-rate for GDP, KRAS remains in an inactive state until the GDP–GTP exchange takes place as a result of signal transduction. This reaction is catalyzed by guanine nucleotide exchange factors (GEFs), which accelerate the exchange by several orders of magnitude [19]. The activation of a cell surface receptor tyrosine kinase (RTK) triggers the phosphorylation of intracellular tyrosine residues that act as docking sites to the SH2 domain of GRB (Growth factor receptor-bound). In turn, GRB can directly (through the SH3 domain) or indirectly (through the adaptor protein Src homology 2-containing tyrosine phosphatase 2—SHP2) bind GEFs, such as SOS1 (Son of Sevenless) [24,25,26]. GEFs associate with GDP-bound KRAS to form a low-affinity complex, either by inserting residues to hinder the GDP phosphate-binding region and/or Mg^2+^ or remodeling switch II to destabilize the guanine nucleotide [27]. Since KRAS has a similar affinity to GTP and GDP (at the picomolar level) [28], the binding of GTP is favored over GDP because the intracellular concentration of GTP is 10 times higher than that of GDP.

Likewise, the intrinsic rate of GTP hydrolysis is very slow [28]. Yet, this basal rate of hydrolysis prevents the constitutive activation of the Mitogen-Activated Protein Kinase (MAPK) pathway. GTPase Activating Proteins (GAPs) increase the pace of this reaction by several orders of magnitude and act to switch off RAS-mediated signaling. GAPs interact with KRAS through their GAP-related domain (GRD), and an arginine (arginine finger) in the GRD induces the conserved glutamine 61 residue in switch II to attack a water molecule and generate a negatively charged hydroxyl ion that breaks the γ-β-phosphoanhydride bond, generating GDP [19,27]. In addition to the arginine finger in GAPs and the glutamine 61 residue in KRAS, the glycine 12 residue in KRAS helps stabilize the complex in the transition state, explaining why mutations in the G12 codon drastically decrease the rate of spontaneous and GAP-stimulated GTP hydrolysis [29]. Not surprisingly, the substitution of the glycine 12 residue by different amino acids results in enzymes with diverse GTP-hydrolysis capabilities, with the KRAS G12C and G12D mutants keeping the highest residual intrinsic and GAP-mediated GTPase activities, respectively [28].

Figure 3 summarizes upstream and downstream members of selected RAS signaling pathways. Effector proteins interact with RAS in the active configuration through their RAS-binding domain (RBD). Each RAS molecule presents one binding site for RBD, such that in a given time, only one effector can bind to RAS [15]. Prediction studies have suggested that the human genome encodes 56 RBD-containing effector proteins [15]. However, not all these effectors bind to RAS with the same affinity. RAF (ARAF, BRAF and CRAF), RALGDS, RGL2, PLCE1, RIN, and RASSF5 are expected to bind with high affinity (nanomolar level) to RAS. Interestingly, binding might be modulated by the concentration of RAS–GTP complexes, tissue-specific RAS and effector abundances, and effector protein recruitment to cellular membranes [15]. These phenomena could potentially explain the pathological role played by other effectors, such as PIK3CA, in the pathogenesis of PDAC. Ultimately, the activation of these effectors leads to signal transduction that activates multiple intracellular pathways, such as RAF/MEK/ERK (MAPK), PI3K/AKT/mTOR, and RalGDS/p38MAPK [30,31].

KRAS signaling pathway activation leads to increased cell proliferation, and additional sources of energy substrates are needed to preserve the mitochondrial function of neoplastic cells, especially in the hypoxic and nutrient-deprived PDAC tumor microenvironment [32]. Therefore, additional mechanisms of nutrient acquisition are necessary. One of these mechanisms, autophagy, seems to be increased both in PDAC and stellate cells [33,34]. Specifically in PDAC cells, autophagy seems to be regulated by a broader transcriptional program involving the MiT/FTE family of transcription factors [35]. In PDAC cells, MiT/FTE proteins are decoupled from their regulatory mediators, and the increased nuclear import of these proteins leads to activation of lysosomal function and nutrient scavenging. Moreover, inhibition of the MAPK signaling pathway induces autophagy due to activation of the LKB1→AMPK→ULK1 axis and increases the dependence of PDAC cells on autophagy [36,37].

## 3. Frequency of *KRAS* Mutations in PDAC

PDAC is the human malignancy most commonly associated with *KRAS* mutations. More than 90% of all PDACs harbor a mutation in *KRAS*. Most mutations are located in the sequence encoding the P-loop (G12 position; 90%) or the switch II region (Q61 position; 7%) [38]. The most common mutations in the G12 position are G12D (36–45% of all mutations), G12V (28–39%), G12R (12–21%), and G12C (0–4%)—Figure 4.

## 4. Prognostic and Predictive Value of *KRAS* Mutation in PDAC

Patients with *KRAS*-mutated PDAC have a worse prognosis when compared to patients with wild-type *KRAS* tumors [39,40]. Also, recent data suggest differences in prognosis based on the type of *KRAS* mutation. In a large dataset of patients with *KRAS*-mutated PDAC, the median overall survival (OS) of patients with *KRAS* G12D-mutated PDAC was significantly shorter when compared to the one of patients with *KRAS* G12R-mutated tumors. There were no differences in survival among patients with tumors harboring mutations in *KRAS* G12R, G12V, or G12C [41,42]. Additionally, preliminary data suggest that the type of *KRAS* mutation might impact the efficacy of first-line chemotherapy. In this same dataset, while patients with tumors harboring *KRAS* G12D and G12V mutations experienced longer survival when treated with FOLFIRINOX, in the small group of patients with *KRAS* G12C-mutated tumors (N = 31), OS was longer in the subgroup treated with gemcitabine plus nab-paclitaxel [43]. Additionally, *KRAS* variant allele frequency (VAF) and *KRAS* allelic imbalance also have prognostic implications. Patients harboring tumors with higher *KRAS* VAF have shorter survival. Also, allelic imbalance leads to increased mutant *KRAS* dosage and is associated with more aggressive clinical behavior [44,45].

## 5. Targeting KRAS

### 5.1. Targeting KRAS Post-Translational Modifications and Trafficking

Given the role of FTase in KRAS plasma membrane anchoring, FTase inhibitors have been developed. However, these drugs have demonstrated no anti-tumor activity in patients with PDAC [46,47,48] despite one anecdotal report of a patient with sustained complete response [49]. Further studies showed that despite the inhibition of FTase, prenylation of KRAS and NRAS, but not of HRAS, continued to be performed by geranylgeranyl transferase (GGTase), explaining the poor response to FTase inhibitors [50]. To address this issue, a dual inhibitor of FTase and GGTase has been developed [51], but further clinical development did not occur.

### 5.2. Targeting KRAS Oligomerization

Plasma membrane-bound KRAS monomers form binary or even higher-order structures (nanoclusters) [19]. While it is not currently known whether the disposition of KRAS in these nanoclusters is needed for its full-blown biological activity [22], this raises the idea that disrupting the interaction between KRAS monomers might lead to lower levels of KRAS signal transduction. Using Monobody technology, a potent RAS biological inhibitor was discovered—NS-1. It binds selectively to the α4–α5 interface of KRAS and HRAS with low nanomolar affinity, inhibiting RAS dimer assembly [52]. In vitro analyses and syngeneic mouse models have shown that NS-1 impairs KRAS signaling, resulting in significant anti-tumor activity [53]. However, recent data suggest the effect of NS-1 is not driven by the disruption of bona fide dimers of RAS but by steric hindrance of the RAS molecule [22]. Therefore, further studies are needed to optimize this strategy in the treatment of PDAC.

### 5.3. Targeting Upstream Components of RAS Signaling Pathways

#### 5.3.1. EGFR

The Epithelial Growth Factor Receptor (EGFR, also known as HER1) is overexpressed in 30–95% of PDACs [54]. However, the addition of the anti-EGFR monoclonal antibodies cetuximab or panitumumab to gemcitabine-based therapy did not lead to improved outcomes in patients with advanced PDAC in randomized controlled trials (RCT) [55,56]. In the NCI-CCTG phase III RCT, the addition of erlotinib (a first-generation small molecule tyrosine kinase inhibitor (TKI) against EGFR) to gemcitabine was associated with statistically significant, albeit not clinically relevant, improvement in median OS in the first-line setting [57]. However, subsequent clinical trials failed to show any significant activity of anti-EGFR TKIs [58,59,60]. The most likely explanation for the lack of activity of isolated anti-EGFR therapy in PDAC is the constitutive activation of intracellular signaling pathways by mutant *KRAS*. Indeed, when tested in patients with wild-type *KRAS* PDAC, the anti-EGFR monoclonal antibody nimotuzumab demonstrated increased efficacy when combined with gemcitabine in two RCTs [61,62].

#### 5.3.2. SOS1

Son of Sevenless 1 (SOS1) is a ubiquitous guanine nucleotide exchange factor (GEF) encoded by the *SOS1* gene. While mammalian cells also contain another related SOS protein (SOS2), its role is currently not well documented. SOS1 interacts with KRAS in response to upstream stimuli by catalyzing the exchange of GDP for GTP, regulating KRAS activation [63,64]. Targeting KRAS with GTP competitive inhibitors faces several challenges (see below) [65]. As a result, one of the strategies to treat *KRAS*-mutated cancers has been the investigation of SOS1 inhibitors, which target the KRAS–SOS1 interface and do not depend on specific *KRAS* mutations for their action. Through experiments in pancreatic cancer cell lines, it has been shown that the inactivation of SOS1 decreases the survival of *RAS*-mutated tumor cells [66].

SOS1 inhibitors are currently being investigated in several *KRAS*-mutated tumor types in phase I clinical studies, both alone and in combination with other drugs. The first drug of this class to reach clinical testing was BI-1701963 (Boehringer Ingelheim). In a phase I first-in-human dose-escalation trial, including 31 patients with *KRAS*-mutated solid tumors (five PDACs) treated with BI-1701963 alone, 23% achieved stable disease, and none had an objective response [67]. The second part of this trial is currently investigating the role of BI-1701963 in combination with trametinib. The rationale for adding a MEK inhibitor in this context is to eliminate the negative feedback in the MEK/ERK signaling pathway caused by SOS1 inhibition [68]. The combination of SOS1 and MEK inhibitors leads to a more sustained pathway inhibition, and trials are underway. More recently, a compound that mediates SOS1 protein degradation has been developed and has also demonstrated promising activity [69].

#### 5.3.3. SHP2

SHP2 (SRC homology region 2–containing protein tyrosine phosphatase 2) is an oncogenic tyrosine phosphatase encoded by the *PTPN11* gene. It is involved in convergent signal transduction downstream of different RTKs and, therefore, regulates multiple intracellular oncogenic signaling pathways, including RAS/RAF/MAPK, JAK/STAT, and PI3K/AKT/mTOR [70,71]. In parallel, SHP2 also plays an important role in controlling immune cell function and the tumor microenvironment [72].

Like SOS1, SHP2 is required for the entire activation of the RAS/RAF/ERK signaling pathway. Indeed, SHP2 participates in cellular signaling through the RAS/RAF/MAPK pathway by activating SOS1-regulated RAS–GTP loading. SHP2 is part of a scaffolding complex that also includes growth factor receptor-bound protein 2 (GRB2), GRB2-associated binding protein (GAB), and SOS1. This complex helps to promote a linkage between an activated growth factor or cytokine receptor and RAS, supporting oncogenic signaling [19,73,74]. Additionally, SHP2 dephosphorylates Sprouty (SPRY), a negative regulator of RAS signaling, therefore abolishingSPRY-mediated inhibition of GRB2/SOS1-dependent RAS activation [75,76].

For several years, targeting the attractive catalytic site of SHP2 was unsuccessful. However, the development of allosteric inhibitors that selectively target SHP2 brought an air of optimism to this field [77]. Currently, at least seven SHP2 inhibitors (TNO155, RMC-4630, JAB-3068, JAB-3312, RLY-1971, ERAS-601, and BBP-398) are being clinically tested in several solid tumors. TNO 155 (Novartis) has been tested in a phase I dose-finding study that included 125 patients with solid tumors, none with the diagnosis of PDAC. No patient experienced an objective response to treatment, and 22% had a stable disease [78]. In the FLAGSHIP-1 phase I dose-escalation study, 28 patients with solid tumors were treated with the SHP2 inhibitor ERAS-601 (including three patients with PDAC) [79]. One patient experienced an objective response to treatment (endometrial carcinoma). Only one patient with PDAC was considered efficacy-evaluable and had disease progression after 10 weeks of treatment. A phase Ib/II trial investigated the role of RMC-4630 (Revolution Medicines, Redwood City, CA, USA) in combination with cobimetinib (a MEK1/2 inhibitor) in 49 individuals whose tumors harbored RAS pathway mutations. Despite having no data until today on the 10 patients with PDAC enrolled in this trial, three out of seven patients with *KRAS*-mutated colorectal cancer with available tumor response evaluation achieved tumor reduction (all with stable disease by RECIST) [80]. In another phase I trial of RMC-4630 as a single agent, 56 participants with solid tumors were enrolled, half of them with non-small cell lung cancer (NSCLC). Again, no data on patients with PDAC have been reported. Among those with NSCLC harboring *KRAS* G12C mutations, a reduction in tumor volume was reported in 43% [81]. Indeed, at least in NSCLC, the activity of SHP2 inhibitors seems to be greater against tumors with *KRAS* G12C mutations [68]. There has been rising interest in combining either an SOS1 or an SHP2 inhibitor with a KRAS G12C inhibitor in patients with *KRAS* G12C-mutated tumors since KRAS G12C inhibitors rely on the GDP-bound state for their activity. Preclinical models have suggested this synergism [82,83]. Combining SOS1 or SHP2 inhibitors with other drugs brings enthusiasm for treating *KRAS*-mutated PDAC, and the future seems promising.

### 5.4. Targeting Downstream Components of RAS Signaling Pathways

#### 5.4.1. RAFs

Currently, available BRAF inhibitors effectively inhibit BRAF monomers, but they cannot be used to treat tumors harboring *KRAS* mutations [84]. These tumors signal through BRAF and CRAF dimers, and it has been shown that these inhibitors paradoxically activate the MAPK pathway by inducing wild-type BRAF dimerization [85]. RAF dimer inhibitors seem less prone to paradoxical activation of the MAPK pathway and are currently under clinical development. In a phase I trial evaluating the pan-RAF dimer inhibitor lifirafenib (BGB-283; BeiGene, Cambridge, MA, USA), six patients with PDAC harboring *KRAS* mutations were enrolled. The best response achieved was stable disease [86]. Belvarafenib (Hanmi Pharmaceutical, Seoul, Republic of Korea), a potent RAF dimer inhibitor, was combined with cobimetinib in a phase I dose-escalating trial that enrolled patients with *RAS* or *BRAF* mutations [87]. Two patients with PDAC were enrolled; however, the short follow-up of these patients prevented the assessment of the activity in PDAC. Other RAF dimer inhibitors are currently under development in other diseases, such as naporafenib (Novartis, Basel, Switzerland/Erasca, San Diego, CA, USA) and exarafenib (Kinnate Biopharma, San Diego, CA, USA), but so far there are no data on their activity against *KRAS*-mutated PDAC.

#### 5.4.2. MEK1/2

Early clinical trials of MEK inhibitors demonstrated limited activity in PDAC. In the first-line setting, the addition of the MEK1/2 inhibitor trametinib to gemcitabine was not associated with improved efficacy [88]. Similar results were seen in the second-line setting. In a small phase II trial, selumetinib (another MEK1/2 inhibitor) was not associated with improved OS when compared with single-agent capecitabine [89]. Furthermore, combining selumetinib with erlotinib was also unsuccessful in PDAC [90].

Preclinical data suggest pancreatic tumors with *KRAS* G12R mutations might be more sensitive to MEK inhibition as they fail to engage with a key KRAS effector—PI3K catalytic subunit p110α [91]. Indeed, these tumors have a higher frequency of co-occurring mutations in the PI3K/AKT/mTOR pathway and depend on the p100γ PI3K subunit to support micropinocytosis [92]. In a phase II clinical trial, eight patients with *KRAS* G12R-mutant PDAC were treated with selumetinib. There were no radiological responses, and the median progression-free survival (PFS) was 3.0 months. However, three patients experienced stable disease for more than six months [93]. Similarly, in another phase II trial, two cohorts of patients were treated with gemcitabine and the MEK inhibitor cobimetinib: one with tumors harboring *KRAS* G12D or G12V mutations and one with tumors with *KRAS* G12R mutation [94]. Among seven patients with tumors harboring *KRAS* non-G12R mutations, all experienced disease progression and death within two months of treatment initiation. On the other hand, among the six patients whose tumors had *KRAS* G12R mutation, one had a partial radiological response, and the disease control rate was 100%. Also, all patients had a ≥50% decrease in CA 19-9 levels after treatment started, and the median PFS of this latter group was 6.0 months. Therefore, the prolonged disease control experienced with MEK inhibitors in this subgroup suggests that MEK inhibition is active against tumors harboring *KRAS* G12R mutations.

#### 5.4.3. ERK1/2

More recently, data on ERK1/2 inhibitors have emerged on the treatment of solid tumors, including PDAC. In a single-arm phase Ib trial, 18 patients with metastatic PDAC were treated with the combination of gemcitabine plus nab-paclitaxel and the ERK1/2 inhibitor ulixertinib [95]. Only five patients completed two cycles of treatment and were evaluable for disease response. One patient achieved a partial response, and two patients had stable disease. For the 15 patients who received all three drugs, median PFS and OS were 5.46 and 12.23 months, respectively. In the HERKULES-1 phase I/IIb trial, 10 patients with pancreatic cancer received the ERK1/2 inhibitor ERAS-007 (Erasca) [96]. Among seven patients with assessable disease, six discontinued treatment within 2 months from the start, mainly because of disease progression. However, one patient whose tumor had a *KRAS* G12V mutation experienced a partial response lasting for approximately 20 weeks. Currently, other EKR1/2 inhibitors, such as MK-8353 (MSD) and LY-3214996 (Eli Lilly, Indianapolis, IN, USA), are under clinical development for patients with solid tumors, including PDAC.

Altogether, these data suggest that inhibitors of upstream and downstream components of KRAS signaling pathways have limited activity as monotherapy. However, there is great potential for synergic combinations with other recently developed drugs.

### 5.5. Direct KRAS Inhibitors

The development of direct RAS inhibitors that compete with GTP has been hampered by some facts [97]. First, RAS has a picomolar affinity for GTP. Second, the intracellular concentration of GTP is several orders of magnitude higher than RAS’s affinity for GTP (roughly 500 nM). Third, RAS lacks a deep or pharmacologically actionable pocket. Lastly, the different RAS mutant isoforms have different GTP-binding sites, complicating the development of pan-RAS inhibitors. However, recent discoveries have revealed molecular vulnerabilities of mutant RAS that have enabled the development of allele-specific and non-specific inhibitors. The main strategies used in these contemporaneous anti-KRAS approaches involve the use of small-molecule inhibitors, molecular degraders, and molecular glues [98].

#### 5.5.1. Allele Specific RAS Inhibitors

##### G12C Inhibitors

The identification of the allosteric switch II binding pocket (S-IIP) adjacent to the mutated cysteine residue in KRAS G12C allowed the development of small inhibitory molecules with the ability to bind specifically to the adjacent S-IIP site and, through covalent binding to the cysteine residue, trap KRAS G12C in its GDP-bound inactive conformation [99,100,101,102].

The pioneering work of Shokat and colleagues in 2013 revealed that the S-IIP site near the mutated cysteine 12 could be targeted by selective KRAS G12C inhibitors. This led to the development of ARS 1620, the first covalent inhibitor tested in vivo in *KRAS* G12C-mutant models [99,103,104]. Subsequent studies focused on the His95 residue present in the cryptic groove of S-IIP, which could facilitate the binding and activity of covalent inhibitors, leading to the discovery of more potent and selective inhibitors, adagrasib (MRTX849) and sotorasib (AMG-510) [104]. Despite the similar mechanism of action, the way these inhibitors bind to S-IIP is distinct. Adagrasib binds to the cryptic pocket involving residues H95, Y96, and Q99, while sotorasib binds to Y96. This partially explains why the two drugs are not completely interchangeable [101,104,105,106].

Adagrasib is an oral small molecule inhibitor, used at a dose of 600 mg BID, that irreversibly and selectively inhibits KRAS G12C (more than 1000 times more effectively than KRAS WT) [107]. The phase II KRYSTAL-1 study evaluated the use of adagrasib in a cohort of 64 previously treated patients with advanced solid tumors harboring *KRAS* G12C mutations (excluding colorectal and NSCLC), including 21 with advanced PDAC, of whom 76.2% had received two or more lines of prior chemotherapy—Table 1 [99]. In the PDAC cohort, the overall response rate (ORR) was 33.3%. With a median follow-up of 16.8 months, the median PFS and OS in the PDAC cohort were 5.4 months and 8.0 months, respectively. In the overall population, the safety profile was favorable, with the most common treatment-related adverse events (TRAE) being low-grade nausea, diarrhea, fatigue, and vomiting. Grade 3 or higher TRAEs occurred in 27% of cases involving fatigue (6.4%) and QT interval prolongation (6.4%). Dose reduction was needed for 39.7% of the patients, and 44.4% temporarily interrupted treatment, but there were no treatment-related discontinuations or deaths. 

Sotorasib is another small molecule that irreversibly binds and inhibits KRAS G12C. Its activity was evaluated in the CodeBreaK phase 1 and 2 studies, which included a total of 38 previously treated patients with advanced *KRAS* G12C-mutated PDAC, of whom 79% had received two or more lines of chemotherapy [108]. With a median follow-up of 16.8 months, the ORR was 21%, with a time to response and duration of response of 1.5 months and 5.7 months, respectively. The median PFS was 4.0 months, and the median OS was 6.9 months, with a 12-month OS rate of 19.6%. Similar to adagrasib, its tolerability profile was favorable, with 16% of grade 3 TRAEs related to diarrhea and fatigue. Dose reductions or interruptions occurred in 13% of the patients, but there were no treatment-related discontinuations or grade 4/5 toxicities.

Other covalent KRAS G12C inhibitors are in different stages of development, including LY3537982, GDC-6036 (divarasib), JNJ 74699157, D-1553, and JDQ443. Recently, safety data from the phase 1 study with LY3537982 in a cohort of 56 previously treated patients with advanced solid tumors were presented, with eight of these patients having PDAC. The ORR achieved in the pancreatic cohort was 42%, without reaching the maximum tolerated dose, with low-grade adverse effects, including diarrhea, constipation, peripheral edema, and nausea [109]. Similarly, in the GO42144 trial, 137 patients with KRAS G12C-mutated solid tumors were treated with the KRAS G12C inhibitor divarasib (formerly GDC-6036). Among seven patients with PDAC, three achieved a partial response and four had stable disease. Again, the drug was associated with mild toxicity, with rare grades 3 to 5 adverse events [110].

Interestingly, comparing the activity of KRAS G12C inhibitors in immunocompetent versus immunodeficient mice, greater activity is observed in the former scenario, suggesting that the pro-inflammatory tumor microenvironment may sustain therapeutic response, opening up opportunities for studies with combination therapies, including immunotherapy [111,112].

Although the development of allele-specific inhibitors has transformed KRAS, once considered undruggable, into a targetable protein, it is noted that responses occur in only about 20–30% of patients with PDAC, and these responses are often partial and not durable [99,108]. Therapeutic resistance can be classified based on the response to inhibitors as primary or intrinsic and secondary or acquired. It can also be classified based on the resistance mechanism as “on-target,” involving the *KRAS* G12C allele itself (cis) or distinct alleles (trans), and “off-target,” which generally involves the activation of other oncogenic pathways bypassing KRAS inhibition [113,114].

The reasons why many *KRAS* G12C-mutated tumors may not respond initially to the inhibitors are not entirely clear, but tumor heterogeneity may play a significant role. Not all tumor cells depend on KRAS G12C for survival, as other alternative mutations, such as those involving *AKT1*, *AKT2*, *PI3KCA*, and *PTEN*, can activate the PI3K–AKT pathway [113]. Adaptive responses to downstream blockade can lead to compensatory upstream stimulation through wild-type RAS and RAF—mediated by RTK activation, which can reactivate the pathway. Preclinical studies with *KRAS* G12C-mutated NSCLC cells exposed to allele-specific inhibitors showed a temporary state of quiescence followed by cell death in some cells, while another group rapidly bypassed the inhibition through upstream stimulation, involving EGFR and AURKA activation, leading to de novo KRAS G12C production [115].

Acquired resistance eventually develops in all patients who initially respond to KRAS G12C inhibitors. In the KRYSTAL-1 study, patients with NSCLC (n = 27), colorectal cancer (n = 10), and appendiceal cancer (n = 1) who received adagrasib were analyzed with pre- and post-treatment samples [116]. Molecular alterations leading to acquired resistance were discovered in 45% of cases, including secondary alterations in *KRAS*, such as G12D/R/W, G13D, Q61H, R68S, H95D/Q/R, Y96C, and *KRAS* G12C allele amplification; 18% showed multiple alterations. Other alterations leading to bypass mechanisms included *MET* amplification, activating mutations in *NRAS*, *BRAF*, *RAF1*, and *FGFR3*, and loss of function of *NF1* and *PTEN*. In the analysis of 43 patients treated with sotorasib (including one patient with PDAC), secondary resistance was observed in 63%, with acquired alterations seen in *KRAS*, *NRAS*, *BRAF*, *EGFR*, *FGFR2*, and *MYC*. Single-cell experiments identified secondary *RAS* and/or *BRAF* mutations in the same cell as *KRAS* G12C, and these cells bypassed inhibition without affecting the target [117].

Examples of “on-target” resistance include the cis-mutation in the *KRAS* G12C itself, leading to the structural alteration Y96D near the S-IIP, preventing the interaction with sotorasib and adagrasib, leading to loss of activity [104,113,114,116,117]. It is noteworthy that, as mentioned above, the two inhibitors have distinct ways of binding to the protein. Other alterations in the S-IIP, such as R68S and H95D, impair the activity of adagrasib but not of sotorasib, suggesting that sotorasib might be effective in patients with these alterations. Additional “on-target” alterations include trans-mutations, bypassing the inhibition site without affecting the binding site, such as *KRAS* G12D, G12V, and G12W, as well as *KRAS* G12C allele amplification [113].

“Off-target” resistance occurs when, despite KRAS G12C inhibition, alternative pathways to KRAS can bypass and activate main canonical pathways, such as BRAF/MEK/ERK and PI3K/AKT/mTOR [113]. Examples include *MET* amplification, which, upon HGF/MET stimulation, activates the AKT pathway and favors the balance of the GTP-activated state, reactivating the oncogenic pathway through NRAS isoforms (Q61L, Q61R, and Q61K). Another example is the *RET* fusion, which can also activate the BRAF/MEK/ERK and PI3K/AKT/mTOR pathways. Other alterations, such as point mutations or deletions in downstream effectors of the MAPK pathway (such as the *BRAF* V600E mutation), can also lead to resistance.

The idea of dual inhibition (KRAS G12C plus RTK inhibitors), exploring EGFR, FGFR, and MET, is being investigated in clinical trials, as one of the main mechanisms of resistance is adaptive feedback involving the activation of other RTKs. Another approach to overcome resistance is to target the signal transducers of these RTKs, such as SHP2 and SOS1. Preclinical data on the combination of KRAS G12C inhibitors with SHP2 inhibitors (SHP099, TNO155) or SOS1 inhibitors (BI3406) have shown promise, leading to ongoing phase 1 and 2 clinical trials [118,119]. Several other combinations are also being studied, including the combination with AKT–PI3K pathway inhibitors, CDK4/6 inhibitors, immune checkpoint inhibitors, DNA repair inhibitors, and even chemotherapy. Next-generation KRAS G12C inhibitors, such as RM-018, bind to cyclophilin A and form trimeric complexes that target the active form of KRAS, maintaining their activity even in the presence of “on-target” resistance mutations like Y96D.

##### G12D Inhibitors

KRAS G12D lacks the reactive Cys12 adjacent to the switch II region, preventing the nucleophilic attack and covalent binding to this amino acid. However, building on the experience with the development of adagrasib, structure-based optimization studies on drug design using non-classical hydrogen bonding and ion pair interactions led to the discovery of MRTX1133 (Mirati Therapeutics, San Diego, CA, USA), a KRAS G12D inhibitor that binds to the S-IIP of GDP-bound KRAS G12D with nanomolar or subnanomolar affinity [120]. It has been shown to bind to KRAS G12D with an affinity 700-fold stronger than to wild-type KRAS. Additionally, MTRX1133 also inhibits the binding of RAF to the active form of KRAS G12D (to a lesser extent), supporting the activity of the molecule regardless of the KRAS conformational state [121]. In initial preclinical studies, similar to KRAS G12C inhibitors, no sign of synergy was found when MRTX1133 was combined with MEK or ERK inhibitors. However, xenograft models suggested increased activity of MRTX1133 when it was combined with the anti-EGFR agents cetuximab and afatinib and the PI3Kα inhibitor BYL-719 [121].

The activity of MRTX1133 has also been demonstrated in immunocompetent mice models, and treatment for 7 days with MRTX1133 has been shown to lead to important changes in the PDAC tumor microenvironment [122]. Increased vascular density, changes in macrophage polarity toward M1 phenotype and cancer-associated fibroblast phenotype, decreased infiltration of myeloid cells, and increased infiltration of CD3+, CD4+, and CD8+ T cell lymphocytes, along with increased CD8+ T cell lymphocyte activation were demonstrated after treatment with MRTX1133. Also, while T cells do not seem necessary for the activity of MTRX1133, preclinical data suggest T cells might augment the extent and duration of the anti-tumor effect. Indeed, the combination of MRTX1133 and checkpoint inhibitors targeting CTLA-4 and/or PD-1 increased the T cell lymphocyte influx and enhanced tumor regression [123]. This is especially important since many murine models developed rapid disease regrowth after MRTX1133 withdrawal, highlighting the need for prolonged treatment or combinations [121]. Interestingly, MRTX1133 causes regression of Pancreatic Intra-Epithelial Neoplasms (PanIN), suggesting that this drug could even be used for chemoprevention [123]. MRTX1133 is currently being evaluated in a phase I/II clinical trial of patients with advanced solid tumors harboring the *KRAS* G12D mutation (NCT05737706). Other KRAS G12D-specific small-molecule inhibitors are at earlier phases of drug development, such as HRS-4642 (Jiangsu Hengrui Medicine, Lianyungang, China; phase I; NCT05533463), BI-KRASG12D (Boehringer Ingelheim, Germany; preclinical), JAB-22000 (Jacobio, Beijing, China; preclinical), and ERAS-4 (Erasca; preclinical) [98].

An additional approach to target KRAS G12D relies on isoform-specific degraders. Proteolysis-targeting chimera (PROTAC) protein degraders are heterobifunctional small molecules consisting of two ligands bound through a link [124]. One binds a protein of interest, such as KRAS, and the other binds an E3 ubiquitin ligase. The binding of the PROTAC to both molecules targets the protein of interest to ubiquitination and subsequent degradation by the ubiquitin-proteasome system. ASP3082 (Astellas Pharma) is a degrader that binds to both KRAS G12D and E3 ubiquitin ligase, fostering KRAS G12D degradation [125]. Preclinical data suggest significant activity against pancreatic cancer in xenograft mouse models of PDAC. It is currently under investigation in a phase I clinical trial (NCT05382559).

An additional approach to target KRAS G12D consists of molecular glues that link KRAS G12D to other cytosolic proteins, interfering with the interaction with effectors proteins and, therefore, blocking signal transduction. RMC-9805 (Revolution Medicines) is an orally-available molecular glue that binds to Cyclophilin A to form binary complexes in cell cytoplasm. Cyclophylin A is a member of the Cyclophilin family, a group of evolutionarily conserved and ubiquitous peptidyl-prolyl isomerases that catalyze the isoconversion of peptyl bonds from trans-form to the cis-form at the proline residue, acting as molecular chaperones [126]. Cyclophilin A is over-expressed in neoplastic tissue, including PDAC [127]. The RMC-9805/Cyclophilin A bi-complex then binds selectively to the KRAS G12D in the GTP-bound active configuration (ON) to form a covalently bound tri-complex. Studies in vitro have shown that upon treatment with RMC-9805, >90% of the KRAS G12D is crosslinked, leading to significant pathway signaling suppression [128]. RMC-9805 has been shown to be active in multiple murine models of PDAC [129]. Moreover, in murine models of KRAS G12D-mutated PDAC, RMC-9805 decreased tumor infiltration by immunosuppressive myeloid cells, increased T cell infiltration, led to increased major histocompatibility complex (MHC) expression and antigen presentation, downregulation of immune checkpoint inhibitors, and increased T Cell Receptor (TCR) repertoire diversity [130]. Importantly, it demonstrated synergistic effects when combined with an anti-PD-1 monoclonal antibody. As a consequence, RMC-9805 is currently being tested in a phase I trial enrolling patients with KRAS G12D-mutated solid tumors (NCT06040541).

##### Inhibitors of Other KRAS Mutant Isoforms

Currently, inhibitors of other KRAS mutant isoforms are under drug development. Recently, Shokat et al. demonstrated α,β-diketoamide and β-lactone moieties are privileged arginine- and serine-reactive functional groups, respectively [131,132]. This has led to the design of drugs inhibiting KRAS G12R and G12S in the GDP-bound (OFF) configuration. However, the development of such inhibitors still faces some challenges. In the specific case of KRAS G12R inhibitors, the low intrinsic and GAP-mediated GTP-hydrolysis activity means that only a very small fraction of KRAS G12R is in targetable GDP-bound form. In the case of the KRAS G12S inhibitor, the current compound presents sub-optimal selectivity (affinity only 5-fold of that of wild-type KRAS). Regarding the *KRAS* G12V mutation, Jacobio Pharmaceuticals has announced the development of a KRAS G12V-specific inhibitor, but data on its in vitro and in vivo efficacy are still pending [133].

#### 5.5.2. Pan-RAS (Non-Specific) Inhibitors

The development of direct inhibitors targeting all conserved RAS proteins (KRAS4A, KRAS4B, NRAS, and HRAS) may have a broader action than that of allele-specific covalent inhibitors. Some examples of compounds tested with pan-RAS properties include 3144, which can bind to the switch-I site and prevent RAS pocket effector activation, and BI-2852 (Boehringer, Ingelheim, Germany), which acts on the DCAI pocket and reduces SOS1 activity, thereby decreasing ERK and AKT activation [84]. One of the major challenges in using these drugs for clinical purposes is their off-target effects on wild-type RAS cells, which might lead to prohibitive toxicity [134].

More recently, a modification of the chemical structure of the covalent KRAS G12C inhibitor BI-0474 (Boehringer Ingelheim) let to the discovery of BI-2865, which has shown promise as a non-covalent pan-RAS inhibitor that impacts nucleotide exchange activity, preventing wild-type KRAS activation, and affecting a wide spectrum of mutated KRAS, with the exception of KRAS G12R and Q61L/K/R [135]. This compound demonstrated in vivo tumor reduction capability without detrimental effects on the animals. Another compound under study is ADT-007 (ADT Pharmaceuticals, Orange Beach, AL, USA), a pan-RAS inhibitor capable of acting on both mutated RAS and wild-type RAS that are constitutively activated due to upstream stimuli. Its mechanism of action involves binding to RAS, blocking GTP binding, interfering with RAS activation, and modulating downstream effectors, as well as the tumor microenvironment. Preliminary data in gastrointestinal models are promising [136].

Other compounds, such as pan-RAS degraders based on macromolecules fused with specific E3 ligases (PROTACs), are also being studied, with interesting in vivo results with specific proteolysis induction of both mutant and wild-type KRAS [97,137]. However, these degraders selectively inhibit proliferation induced only by the *KRAS*-mutant allele [137]. The pan-RAS inhibitor RMC-6236 (Revolution Medicines, Redwood City, CA, USA) is currently being tested in a phase I trial enrolling patients with tumors harboring KRAS G12D, G12A, G12R, G12V, G12S mutations (NCT05379985).

## 6. Targeting Metabolic Reprogramming

Autophagy is a highly conserved physiological cellular process involved in the catabolism of damaged proteins and organelles. Under stressful conditions, autophagy is enhanced to degrade intracellular compounds and recycle macromolecule precursors to preserve cellular turnover [138,139]. The commonly known mechanism of autophagy (defined as macroautophagy) involves the engulfment of a portion of the cell cytosol in vesicles (autophagosomes) that eventually fuse with lysosomes, leading to degradation and posterior formation of new metabolites [140]. This process involves at least seven steps and is governed by a series of autophagy-related genes (ATGs) [140,141].

Autophagy is a complex and not yet fully understood biological phenomenon. It has been considered a double-edged sword since autophagy dysregulation may both suppress and promote tumorigenesis. High levels of autophagy are seen in primary pancreatic tumors, given their usual hypoxic and nutrient-deprived environment [142]. In PDAC, the activation of autophagy may lead to epithelial-mesenchymal transition (EMT), regulation of energy homeostasis, as well as mediation of chemotherapy-induced resistance [142]. Therefore, autophagy inhibition may be explored as a potential therapeutic strategy. In murine models in which tumor cells had one of the autophagy genes deleted, despite the increased incidence of premalignant lesions (such as PanINs), the growth of neoplastic cells was halted [32,143,144].

Interestingly, autophagy is upregulated and has proved to be essential for oncogenic *KRAS*-induced malignant cell transformation [145]. Since *KRAS* mutations occur in approximately 90% of PDACs, it makes even more sense to investigate the role of autophagy inhibitors in this type of tumor. Indeed, responses were observed in a mouse model using patient-derived xenografts (with *KRAS*-mutated PDAC) treated with hydroxychloroquine (HCQ), an autophagy inhibitor [146]. Both chloroquine (CQ) and HCQ may inhibit autophagy by blocking the fusion of autophagosomes with lysosomes [147].

In one initial phase I/II trial, 35 patients were preoperatively treated with HCQ and gemcitabine. Interestingly, 61% had a decrease in CA19-9 levels, 94% underwent surgical resection (77% of them R0) and the median OS was very promising at 34.8 months [148]. Another phase I/II single-arm study evaluated the combination of neoadjuvant HCQ and gemcitabine in patients with high-risk resectable PDAC [149]. The median OS was 31 months and 31% of patients survived beyond 5 years, which again compares very favorably with historical data from patients treated with neoadjuvant gemcitabine alone.

However, data from more robust studies suggest HCQ has minimal activity against PDAC. In a phase II single-arm trial, 20 patients with previously treated advanced PDAC were given HCQ at two dose levels (400 mg or 600 mg twice a day). No objective responses were seen and only two patients (10%) were progression-free at 2 months. Median PFS and OS were 46.5 and 69 days, respectively. Additionally, HCQ achieved inconsistent autophagy inhibition as measured by the autophagy marker LC3-II in peripheral lymphocytes [150]. Also, a phase II trial randomized 112 patients with treatment-naïve advanced PDAC to treatment with gemcitabine plus nab-paclitaxel with or without HCQ [151]. Despite the fact that patients assigned to HCQ experienced a higher ORR (38.2 vs. 21.1%; *p* = 0.047), no differences were seen in median PFS (5.7 vs. 6.4 months; *p* = 0.25) or OS (11.1 vs. 12.1 months; *p* = 0.53). These results are in line with those of a phase II trial that randomized patients with potentially resectable PDAC to neoadjuvant gemcitabine plus nab-paclitaxel with or without HCQ [152]. Again, the authors described improved (pathological) response for those treated with HCQ, although there were no differences in relapse-free survival and OS. A subsequent analysis of this study suggested tumors with SMAD4 loss had a better response to treatment with HCQ. Nonetheless, further data on the role of other key co-occurring mutations in the setting of autophagy are needed [153].

One important issue raised by some authors is that given that HCQ is a relatively weak inhibitor of autophagy (demanding micromolar levels to inhibit autophagy in patients) [154], high doses should be administrated in order to achieve adequate outcomes [155]. That might partially explain some of the negative results seen so far.

In vitro and in vivo models have recently demonstrated that there is a common compensatory increase in IGFR1 when there is exposure to HCQ [156]. Additionally, inhibition of IGFR1 leads to enhanced autophagic flux and sensitivity to HCQ. Importantly, MEK inhibitors increase the metabolic dependency of cancer cells on autophagy in the presence of *KRAS* mutations. Indeed, this has been supported by preclinical data [36,37] and case reports [36,155] suggesting that the combination of MEK inhibitors and HCQ might be active in chemo-refractory PDAC. As a result, several clinical trials investigating the combination of HCQ with the MEK inhibitors trametinib (NCT03825289), cobimetinib (NCT04214418), and binimetinib (NCT04132505), and with the EKR inhibitor ulixertinib (NCT04145297) are currently enrolling patients with advanced PDAC.

Autophagy also plays a fundamental role in the development, maturation, and modulation of several immune cells [157]. More recently, it has been shown that autophagy also promotes immune evasion of PDAC by degrading MHC class I through autophagy cargo receptor NBR1, leading to impaired antigen presentation. In mice models, inhibition of autophagy can restore surface levels of MHC class I and enhance anti-tumor T cell responses [158]. Therefore, a potential synergy of immunotherapy and autophagy inhibitors has been speculated.

To better target autophagy, different technologies have been applied to CQ and HCQ. Chloroquine-loaded nanoparticles and polymeric HCQ (a macromolecular multivalent version thereof) have been explored with promising data [158,159]. In addition to them, many other autophagy inhibitors targeting lysosomes are in clinical development, such as the Lys05 family, GNS561 (Genoscience Pharma, Marseille, France), and ROC-305 [160]. DCC-3116 (Deciphera Pharmaceuticals, Waltham, MA, USA) is an inhibitor of the ULK1/2 kinases, which initiate autophagy in response to the blockade of the MAKP and PI3K pathways [161]. It is currently being evaluated in a clinical trial that is enrolling patients with tumors harboring mutations in the MAPK pathway. MicroRNAs may also serve as an alternative approach for autophagy inhibition as they may target ATGs [162]. Therefore, the future seems promising for autophagy inhibitors, especially when combined with other agents.

## 7. Targeting KRAS and Co-Occurring Genetic Alterations

### 7.1. CDKN2A

*CDKN2A* codifies two tumor suppressor proteins responsible for cell cycle arrest in the G1/S checkpoint. p16 inhibits the binding of CDK4/6 to cyclin D, leading to G1/S cell cycle arrest and ARF14 participates in p53-dependent cell cycle arrest [163]. Given the mutation pattern of CDKN2A/B, p16 seems to be the most important transcript in the pathogenesis of PDAC. Roughly 90% of PDACs present CDKN2A inactivation, making it an important therapeutic target. However, both in molecularly non-selected and selected patients, CDK4/6 inhibitors failed to demonstrate significant activity against PDAC [164,165,166]. Preclinical studies suggest that a combination of MEK and CDK4/6 inhibitors might lead to cancer cell senescence [167]. Indeed, in one small retrospective analysis of patients with tumors harboring co-occurring RAS/RAF/MEK and cyclin gene alterations, six patients with PDAC were treated with the combination of trametinib and the CDK4/6 inhibitor palbociclib. Two patients experienced partial response with a PFS of 9 and 17.5+ months [168]. This combination is currently being tested in the EAY191-3 substudy (cohort) of the comboMATCH initiative [169]. Preclinical data also suggest that the combination of ERK and CDK4/6 inhibitors might lead to increased levels of apoptosis in PDAC [170]. Clinical trials are currently evaluating this combination for patients with advanced PDAC (NCT03454035, NCT04870034). However, in one initial investigation, the ERK1/2 inhibitor ulixertinib was given in combination with palbociclib to 26 patients with solid tumors (including nine with PDAC). Despite the short median PFS in the overall population (2.5 months), one patient with PDAC experienced a prolonged benefit [171]. Similarly, in the HERKULES-3 phase I trial, 24 patients with PDAC were treated with the combination of the ERK1/2 inhibitor ERAS-007 (Erasca) and palbociclib. No patient experienced radiological response and the median PFS was less than 10 weeks [172]. Interestingly, a recent study using CRISPR (see below) suggests that CDK2, a cyclin-dependent kinase, might compensate for CDK4/6 inhibition and is a potential target in combination with ERK1/2 and CDK4/6 inhibitors [170].

### 7.2. PIK3CA/AKT/mTOR

Both the RAF/MEK/ERK and the PI3K/AKT/mTOR signaling pathways are activated by RAS and share feedback mechanisms that provide significant crosstalk between these two pathways [173]. This was supported by initial studies showing the synergistic effect of the combination of MEK and PI3K/AKT inhibitors in preclinical PDAC models [174]. However, clinical trials evaluating this combination failed to show significant benefits. In a phase I trial, the AKT inhibitor GSK2126458 was given in combination with the MEK inhibitor trametinib to 69 patients with solid tumors, including seven patients with PDAC [175]. Among these patients, the best response was stable disease in two patients. In a randomized phase II trial, 120 patients experiencing disease progression during gemcitabine-based first-line therapy were randomized to treatment with mFOLFOX or the AKT inhibitor MK-2206 in combination with selumetinib. Patients in the FOLFOX arm experienced improved OS and PFS. Among 58 patients included in the MK-2206+ selumetinib arm, one patient experienced a radiological response [176]. Similar results have been demonstrated with the combination of trametinib with the AKT inhibitor afuresertib [177]. In a dose–escalation phase I trial evaluating the combination of the pan-class I PI3K inhibitor buparlisib and the MEK inhibitor binimetinib in patients with solid tumors harboring alterations in RAS/RAF, seven patients with PDAC were enrolled [178]. None of them experienced radiological response. More recently, Ciuffreda and colleagues failed to demonstrate the activity of MEK and PI3K inhibitors in preclinical models of PDAC, except for those with inactivation of PTEN, potentially explaining the lack of clinical efficacy observed in the clinical trials of combined MEK and PI3K/AKT inhibitors [179].

## 8. KRAS Vaccine

*KRAS*-mutated proteins present neoantigens with high immunogenicity potential, and vaccines could offer a very specific anti-tumor effect. Such vaccines are composed of long synthetic KRAS-mutated peptides and aim to enhance T-helper lymphocyte responses and memory. However, preclinical data showed limited efficacy in monotherapy. This could be partially explained by the fact that single peptides fail to produce adequate epitopes for the T-cell activation process [180].

The results of initial investigations in this field were conflicting. An American phase I/II trial with 24 patients with *KRAS*-mutated PDAC evaluated the effects of a vaccine against the specific KRAS G12-mutated peptides as adjuvant therapy after complete resection [181]. Specific immune response was detected in only one patient, while three patients had non-specific responses. Despite mild toxicity, clinical efficacy was also limited. On the other hand, a European phase II trial with 20 patients reported 17 immunological responses (not described as specific or not by the authors). With long-term follow-up, this study showed a 7% difference in 5-year OS rate between the whole population and patients with immunological response to the vaccine [182]. Immune responses seem to be further improved using adjuvants, such as intradermal granulocyte-macrophage colony-stimulating factor (GM-CSF). In a phase I study, 25 out of 43 patients treated with synthetic RAS-mutated peptides and GM-CSF showed signs of immune response to the vaccine and these patients experienced improved survival [183].

Despite these promises, the use of peptide-based vaccines against KRAS has yet to show practice-changing clinical activity. A recent randomized phase II study evaluated the role of the yeast-based vector vaccine GI-4000 in the adjuvant setting. After surgery, patients were randomized to receive gemcitabine with either GI-4000 or placebo. Forty percent of patients achieved an immune response to the vaccine. However, the study failed to show significant differences in median PFS or OS [184].

With the COVID-19 pandemic, renewed interest in the field of vaccines against PDAC has emerged, especially with messenger RNA (mRNA) vaccines. mRNA vaccines do not integrate into the genome or generate a humoral response. After taking up the complex vaccine–KRAS, APCs stimulate only a cytotoxic T-cell response. Nonetheless, this mechanism generates more memory T-cell responses with time [185]. An anti-KRAS mRNA vaccine (mRNA-5671/V941) is currently being studied in combination with pembrolizumab in patients with tumors harboring *KRAS* mutations, including PDAC (NCT 03948763).

## 9. Interfering with mRNA

Given the difficulties in targeting KRAS, new approaches to tackle this problem have been designed. An interesting concept is to interfere with the translation of the *KRAS*-mutated mRNA. A first attempt used specific ribozyme-mediated reversal of malignant phenotype in preclinical models of PDAC. Despite tumor regression or suppression in a significant number of mice, this approach was not further developed [186].

An alternative way is inhibition at the transcriptional level, interfering with the mRNA function. Its principal limitations are the specificity of the mutant KRAS and the repercussions in the wild-type RNA transcription. A technology capable of solving this problem is the microbial Clustered Regularly Interspaced Short Palindromic Repeats (CRISPR) technique, which is commonly used for gene editing. It can produce enzymes guided to target a specific effector with a complementary sequence. In mammalian models, it has been shown to knockdown mutant-specific RNA without affecting the wild-type version [187]. In PDAC, preclinical studies using CRISPR–Cas13 and CRISPR–Cas9 enzymes that specifically inhibit KRAS–G12D mRNA showed a significant reduction in signal transduction by KRAS-mutated proteins while not affecting wild-type KRAS activity [188,189].

The use of small interfering RNA (siRNA) is also being tested. Preclinical data show that the exosomes derived from fibroblast-like mesenchymal cells engineered to carry siRNA can target oncogenic KRAS G12. This effect is dependent on CD47 and is facilitated by micropinocytosis [190]. A phase I study is currently enrolling patients with advanced PDAC for treatment with mesenchymal stromal cells-derived exosomes with KRAS G12D siRNA (NCT03608631).

## 10. Conclusions

Despite the molecular heterogeneity, *KRAS* mutations are almost universal in PDAC, playing a critical role in its carcinogenesis. Considered an undruggable target for many decades due to its conformational characteristics and high affinity for GTP, KRAS protein has recently been targeted by small inhibitory molecules that bind specifically to the allosteric switch II binding pocket in KRAS G12C, trapping the protein in its inactive conformation. This achievement has ushered a new era of therapeutic development in PDAC. Preliminary clinical trials have demonstrated promising findings, but frequently with partial and short-term responses. Primary and secondary mechanisms of resistance to KRAS G12C inhibitors have been intensively investigated. Results of clinical trials evaluating inhibitors of other isoforms of KRAS and exploring combinations of KRAS G12C inhibitors with monoclonal antibodies and with receptor tyrosine kinase inhibitors are eagerly awaited for the next few years—Table 2. The recent advances in the therapeutic development of RAS inhibitors associated with other therapeutic strategies that have been explored in PDAC and in other solid tumors, such as immunotherapeutic approaches and adoptive T-cell therapies, bring optimism and hope for the increasing number of patients affected by this extremely complex and challenging disease.

## Figures and Tables

**Figure 1 cancers-15-05015-f001:**
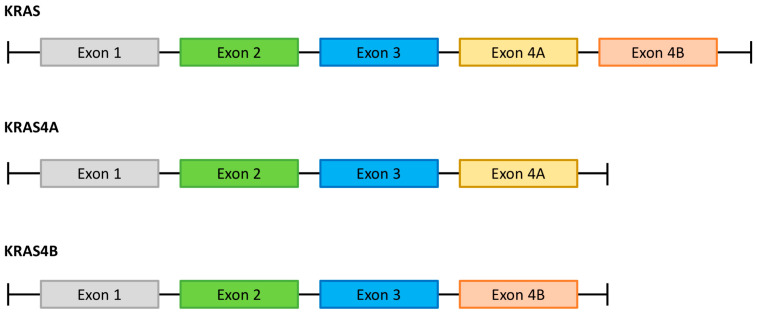
Alternative splicing of the *KRAS* gene. The existence of the two KRAS isoforms (KRAS4A and KRAS4B) is a consequence of alternative splicing of the last exon (exon 4) of the *KRAS* gene.

**Figure 2 cancers-15-05015-f002:**
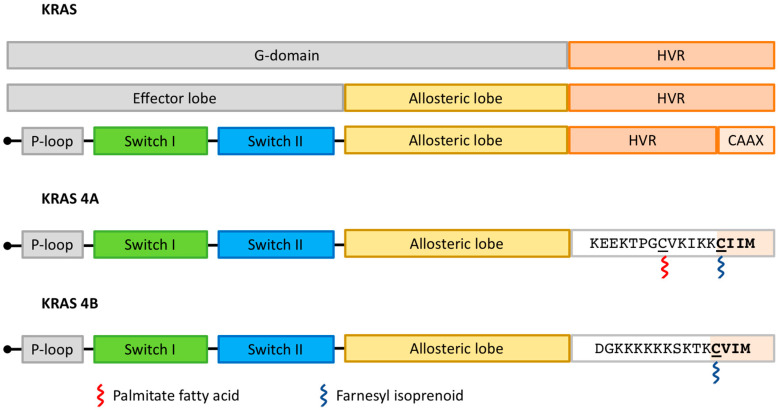
Functional domains of the KRAS isoforms and docking sites of prenylation. Both KRAS4A and 4B isoforms consist of two functional parts: a G domain and a Hypervariable Region (HRV). The G domain is further divided into effector and allosteric lobes. Key enzymatic sites are located in the effector lobe, such as the P-loop (amino acids 10 to 17), switch I (amino acids 30–38), and switch II (amino acids 59–76). Sites for KRAS prenylation are located in the HVR and do not completely overlap between KRAS isoforms. HVR: Hypervariable Region; CAAX: C means cysteine, A means aliphatic amino acid, and X means any amino acid.

**Figure 3 cancers-15-05015-f003:**
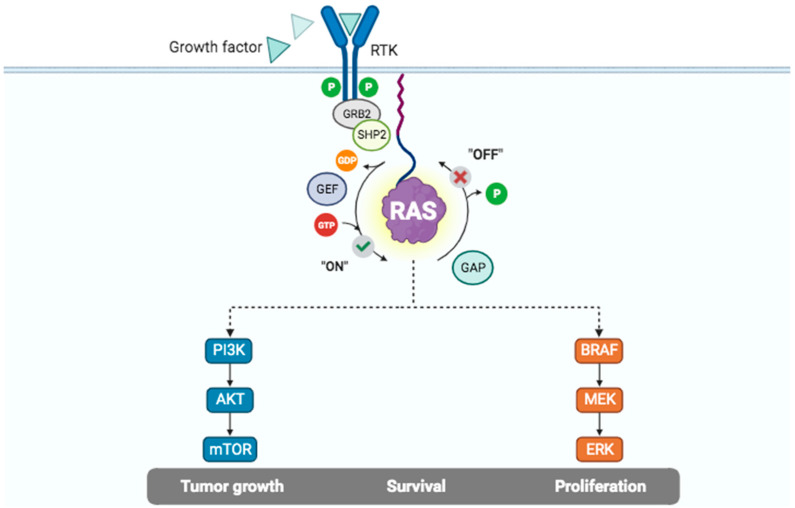
RAS function and intracellular signaling pathways. Summary description of RAS signaling pathways. Upon binding of growth factor to the Receptor Tyrosine Kinase (RTK), phosphorylation of specific residues in the intracellular component of the RTK drives adaptor proteins (such as GRB2 and SHP2) that activate RAS-guanine nucleotide exchange factors (RAS-GEFs, such as SOS1). This increases in several orders of magnitude the formation of RAS–GTP, which is the active form of the enzyme. Apart from spontaneous hydrolysis, RAS-bound GTP hydrolysis in mediated by RAS–GTPase activating proteins (GAPs, such as NF1), leading to inactivation of RAS. Activation of RAS leads to downstream activation of multiple effector pathways (such as PI3K/AKT/mTOR and the MAPK pathways). For further details, please see the manuscript by Nissley DV and McCormick F [25]. Figure created with BioRender 2023.

**Figure 4 cancers-15-05015-f004:**
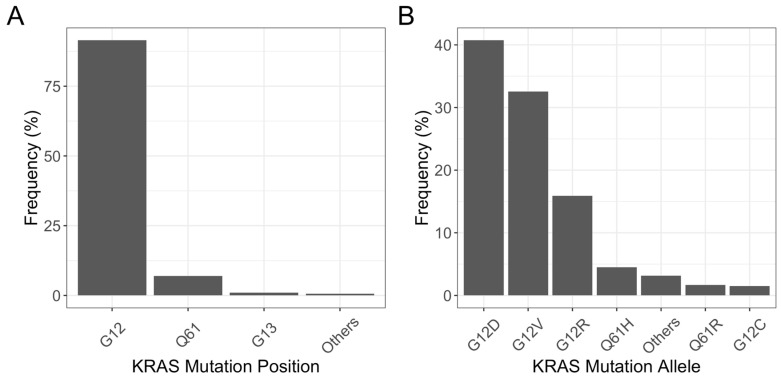
*KRAS* mutation position (**A**) and mutant *KRAS* allele (**B**) in patients with PDAC (number of samples= 1552) [38]. Data extracted from cBioPortal on 25 July 2023 using the following datasets: Pancreatic Adenocarcinoma (ICGC, Nature 2012), Pancreatic Adenocarcinoma (TCGA, PanCancer Atlas), Pancreatic Cancer (UTSW, Nat Commun 2015), Pancreatic Adenocarcinoma (QCMG, Nature 2016), MSK-IMPACT Clinical Sequencing Cohort (MSK, Nat Med 2017), Pancreatic Ductal Adenocarcinoma (CPTAC, Cell 2021), China Pan-cancer (OrigiMed, Nature 2022). Filters included: “Pancreatic Cancer” and “Pancreatic Adenocarcinoma”. The Other categories include KRAS Mutation Positions and KRAS Mutation Alleles with a frequency of less than 1%.

**Table 1 cancers-15-05015-t001:** Activity and safety profile of KRAS G12C specific inhibitors in advanced PDAC.

Drug	Adagrasib (MTRX 849)	Sotorasib (AMG 510)
Dose	600 mg twice a day	960 mg once a day
Half-life (hours)	23	5.5 ± 1.8
Study	KRYSTAL-1	CodeBreaK100
Phase	II	I/II
Sample size	21	38
ORR (%)	33.3	21.0
Median TTR (months)	1.4 ^#^	1.5
DCR (%)	81.0	84.0
DOR (%)	5.3 ^#^	5.7
Median PFS (months)	5.4	4.0
Median OS (months)	8.0	6.9
Grades 3–4 toxicities (%)		
Fatigue	6.3 ^#^	2.3
AST increase	3.2 ^#^	2.3
ALT increase	-	4.7
QT prolongation	6.3 ^#^	-
Diarrhea	1.6 ^#^	3.9
Vomiting	1.6 ^#^	3.9
Anemia	1.6 ^#^	4.7

^#^ For the overall population of patients with solid tumors. ORR: Overall Response Rate; TTR: Time to Response; DOR: Duration of Response; PFS: Progression-free Survival; OS: Overall Survival; AST: Aspartate Aminotransferase; ALT: Alanine Aminotransferase.

**Table 2 cancers-15-05015-t002:** Currently ongoing clinical trials evaluating different strategies against *KRAS*-mutated PDAC.

Drug	Setting	Phase	Clinical Trial Identifier
KRAS Inhibitors—G12C			
HBI-2438	Advanced solid tumors with *KRAS* G12C mutation	I	NCT05485974
Adagrasib	Advanced PDAC with KRAS G12C mutation	I	NCT05634525
RMC-6291	Advanced solid tumors with *KRAS* G12C mutation	I	NCT05462717
BPI-421286	Advanced solid tumors with *KRAS* G12C mutation	I	NCT05315180
D-1553 (Garsorasib)	Advanced solid tumors with *KRAS* G12C mutation	I	NCT04585035
LY3537982	Advanced solid tumors with *KRAS* G12C mutation	I	NCT04956640
BI 1,823,911 (alone or in combination with other agents)	Advanced solid tumors with *KRAS* G12C mutation	I	NCT04973163
JAB-21822	Advanced solid tumors with *KRAS* G12C mutation	I/II	NCT05002270
JNJ-74699157	Advanced solid tumors wih *KRAS* G12C mutation	I	NCT04006301
JDQ443 (alone or in combination with other agents)	Advanced solid tumors with *KRAS* G12C mutation	I	NCT04699188
LY3537982 (alone or in combination with other agents)	Advanced solid tumors with *KRAS* G12C mutation	I/II	NCT04956640
MK-1084 (alone or in combination with other agents)	Advanced solid tumors with *KRAS* G12C mutation	I	NCT05067283
Adagrasib (alone or in combination with other agents	Advanced solid tumors with *KRAS* G12C mutation	I/II	NCT03785249
Sotorasib (alone or in combination with other agents)	Advanced solid tumors with *KRAS* G12C mutation	Ib/II	NCT04185883
**KRAS inhibitor—G12D**			
MTRX1133	Advanced solid tumors with *KRAS* G12D mutation	I/II	NCT05737706
HRS-4642	Advanced solid tumors with *KRAS* G12D mutation	I	NCT05533463
RMC-9805	Advanced solid tumors with *KRAS* G12D mutation	I	NCT06040541
**Pan-RAS inhibitor**			
RMC-6236	Advanced solid tumors with *KRAS* mutation (except G12C)	I/Ib	NCT05379985
**KRAS degrader**			
ASP3082	Advanced solid tumors with *KRAS* G12D mutation	I	NCT05382559
**SOS1 inhibitor**			
MRTX0902 (alone or in combination with adagrasib)	Advanced solid tumors with mutations in MAPK genes	I/II	NCT05578092
BI 1701963 (alone or in combination with trametinib)	Advanced solid tumors with *KRAS* mutation	I	NCT04111458
**SPH2 inhibitor**			
RMC-4630	Advanced solid tumors with MAPK alteration	I	NCT03634982
RMC-4630 + LY3214996	Advanced solid tumors with *KRAS* mutation	I/Ib	NCT04916236
JAB-3068	Advanced solid tumors	I/IIa	NCT03565003
JAB-3312	Advanced solid tumors	I	NCT04121286
JAB-3312	Advanced solid tumors	I	NCT04045496
GDC-1971 + Atezolizumab	Advanced solid tumors	I	NCT05487235
**KRASi + anti-EGFR**			
JAB-21822 + Cetuximab	Advanced solid tumors with *KRAS* G12C	Ib/II	NCT05194995
**KRASi + SOS1i**			
Adagrasib + BI 1701963	Advanced solid tumors with *KRAS* G12C	I/Ib	NCT04975256 (KRISTAL-14)
**KRASi + SPH2i**			
Adagrasib + TNO155	Advanced solid tumors with *KRAS* G12C	I/II	NCT04330664 (KRISTAL-2)
JAB-21822 + JAB-3312	Advanced solid tumors with *KRAS* G12C	I/IIa	NCT05288205
**Autophagy inhibitors**			
DCC-3116 (ULK1/2 inhibitor)	Advanced solid tumors with *RAS*/*BRAF* mutation	I/II	NCT04892017
Hydroxychloroquine + Trametinibe	Advanced PDAC	II	NCT05518110 (PaTcH)
Hydroxychloroquine + Trametinibe	Advanced PDAC	I	NCT03825289 (THREAD)
Hydroxychloroquine + Binimetinib	Advanced PDAC with *KRAS* mutation	I	NCT04132505
**MEK inhibitors**			
IMM-1-104	Advanced solid tumors with *RAS* mutation	I/IIa	NCT05585320
**Adoptive cell therapy**			
Mutant KRAS G12V-specific TCR transduced T cell therapy	Advanced PDAC with *KRAS* G12V mutation	I/II	NCT04146298
Mutant KRAS G12V-specific TCR transduced T cell therapy	Advanced PDAC with *KRAS* G12V mutation	I/II	NCT03190941
Mutant KRAS G12D-specific TCR transduced T cell therapy	Advanced PDAC with *KRAS* G12D mutation	I/II	NCT03745326
**Vaccine**			
ELI-002	Solid tumors with mutations in *KRAS*/*NRAS* (G12R/D) and minimal residual disease (ctDNA or CA 19-9 +)	I/II	NCT05726864 (AMPLIFY-7P)
ELI-002	Solid tumors with mutations in *KRAS*/*NRAS* (G12R/D) and minimal residual disease (ctDNA or CA 19-9 +)	I	NCT04853017 (AMPLIFY-201)
mRNA-5671/V941 (alone or in combination with Pembrolizumab)	Advanced solid tumors with *KRAS* mutation	I	NCT03948763 (V941-001)
Mutant KRAS-Targeted Long Peptide Vaccine Combined + Nivolumab + Ipilimumab	Resected PDAC with *KRAS* mutation	I	NCT04117087
**Small interfering RNA**			
iExosomes	Advanced PDAC with *KRAS* G12D mutation	I	NCT03608631

## Data Availability

All data included in this manuscript can be provided upon reasonable request.

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
