# Peer review of "Targeting KRAS in Pancreatic Ductal Adenocarcinoma: The Long Road to Cure"

_cancers, 2023, doi:10.3390/cancers15205015_

Round 1

Reviewer 1 Report

  1. In fact, there have been several reviews examining the role of KRAS in PDAC and discussing strategies for targeting it. Much of the content in this review draws from existing reviews (e.g., reference: PMC8556325), and it would be beneficial for the author to clarify this in the introduction.

  2. While the author delves into strategies for targeting metabolic reprogramming, it might be helpful to provide background information on the relationship between KRAS and metabolic reprogramming in the preceding section.

  3. Have novel approaches for targeting KRAS in PDAC, such as PROTACs and α4–α5 interface inhibitors, been considered?

  4. Are there any innovative combination strategies for targeting KRAS in PDAC that could be explored?

None

Author Response

In fact, there have been several reviews examining the role of KRAS in PDAC and discussing strategies for targeting it. Much of the content in this review draws from existing reviews (e.g., reference: PMC8556325), and it would be beneficial for the author to clarify this in the introduction.

ANSWER: Thank you for your comment. We added an introduction part and included a short description of the methodology and cited the review articles as suggested.

While the author delves into strategies for targeting metabolic reprogramming, it might be helpful to provide background information on the relationship between KRAS and metabolic reprogramming in the preceding section.

ANSWER: We appreciate this insight. We added a section on the relationship of KRAS and autophagy in the preceding section. 

Have novel approaches for targeting KRAS in PDAC, such as PROTACs and α4–α5 interface inhibitors, been considered?

ANSWER: Thank you for your comment. Indeed, we included the discussion on PROTACs in the KRAS G12D specific inhibitors section and also in the Pan-RAS inhibitors section. I agree that α4–α5 interface inhibitors are promising targets and therefore, we included a paragraph on the potential role of this class of compounds.

Are there any innovative combination strategies for targeting KRAS in PDAC that could be explored?

ANSWER: Thank you for raising this issue. While we did not write a separate section on potential combination therapies, throughout the text, we exemplify potentially synergistic combinations that have putative biological or demonstrated preclinical activity. 

Reviewer 2 Report

The paper „Targeting KRAS in Pancreatic Ductal Adenocarcinoma: the Long Road to Cure” is a review on molecular basis of pancreatic ductal adenocarcinoma (PDAC) development and methods of treatment, including the latest. And this comprehensive description of the methods is the greatest value of this work, which - without it - would be just another review about PDAC. So if the Cancers editors believe that there is still a need for papers on this subject, then it is certainly worth publishing - with minor corrections.

First of all, the dimensions of the objects presented in some illustrations should be changed. Figure 3 occupies a large area of the page, but most of it is empty space, and the main elements of the drawing are small, the font is too small and therefore not very legible. Similarly, the charts in Figure 4 are mostly empty space. Presentation of “gapped” Y axis would improve the readability of data with low values (and if the authors believe that the data is not very important, they could summarize it in one sentence, instead of presenting charts in which most of the Y values are equal 0 – or so it seems).

In some chapters, e.g. “Allele specific RAS inhibitors”, the authors use citations very sparingly and provide long strings of information without clear references to the sources.

Authors should also read their work carefully and correct very minor linguistic errors ("in" instead of "is" – line 89, etc.) and re-write some sentences (e.g. “The main mechanisms used in these contemporaneous anti-KRAS approaches are the use of small-molecule inhibitors, molecular degraders, and molecular glues.” - Can the use of inhibitors be called “a mechanism”?

Author Response

The paper „Targeting KRAS in Pancreatic Ductal Adenocarcinoma: the Long Road to Cure” is a review on molecular basis of pancreatic ductal adenocarcinoma (PDAC) development and methods of treatment, including the latest. And this comprehensive description of the methods is the greatest value of this work, which - without it - would be just another review about PDAC. So if the Cancers editors believe that there is still a need for papers on this subject, then it is certainly worth publishing - with minor corrections.

ANSWER: Thank you for your comments

First of all, the dimensions of the objects presented in some illustrations should be changed. Figure 3 occupies a large area of the page, but most of it is empty space, and the main elements of the drawing are small, the font is too small and therefore not very legible. Similarly, the charts in Figure 4 are mostly empty space. Presentation of “gapped” Y axis would improve the readability of data with low values (and if the authors believe that the data is not very important, they could summarize it in one sentence, instead of presenting charts in which most of the Y values are equal 0 – or so it seems)

ANSWER: Thank you for your comment. We have reframed the picture in an attempt to reduce empty spaces. We also increased the size of some of the elements in the graph to improve visualization. Also, we grouped KRAS mutation positions and alleles with less than 1% frequency into “Others” groups in an attempt to summarize information. 

In some chapters, e.g. “Allele specific RAS inhibitors”, the authors use citations very sparingly and provide long strings of information without clear references to the sources.

ANSWER: Thank you for your comment. We have added new references to this part in an attempt to improve the accuracy of our citations. However, please bear in mind that in some paragraghs, such as the ones describing the studies KRYSTAL-1 and CodeBreak, there are long strings of text without references because we described extensively the results of these studies.

Reviewer 3 Report

It was my pleasure to read and review this manuscript. It is an important summary of the targeting of KRAS in pancreatic cancer. It is a thorough assessment of the current field. Well done!

One very small comment/addition for the authors to consider: there is a new KRAS G12D inhibitor that was not mentioned that is coming into clinical trials right now called RMC-9805 (NCT06040541).

Author Response

It was my pleasure to read and review this manuscript. It is an important summary of the targeting of KRAS in pancreatic cancer. It is a thorough assessment of the current field. Well done!

ANSWER:  Thank you for your comment.

One very small comment/addition for the authors to consider: there is a new KRAS G12D inhibitor that was not mentioned that is coming into clinical trials right now called RMC-9805 (NCT06040541).

ANSWER: In the KRAS G12D-specific inhibitors section, we have discussed the pre-clinical data of RMC-9805. We have added both in the text and in Table 2 that this compound is currently being tested in a phase I trial.

Authors should also read their work carefully and correct very minor linguistic errors ("in" instead of "is" – line 89, etc.) and re-write some sentences (e.g. “The main mechanisms used in these contemporaneous anti-KRAS approaches are the use of small-molecule inhibitors, molecular degraders, and molecular glues.” - Can the use of inhibitors be called “a mechanism”?

ANSWER: Thank you for your advice. We have proofread the paper and corrected these mistakes.

Round 2

Reviewer 1 Report

The quality of the manuscript is improved after revision, and it could be accepted for publication.